# The Papain-like Protease Domain of Severe Acute Respiratory Syndrome Coronavirus 2 Conjugated with Human Beta-Defensin 2 and Co1 Induces Mucosal and Systemic Immune Responses against the Virus

**DOI:** 10.3390/vaccines12040441

**Published:** 2024-04-19

**Authors:** Byeol-Hee Cho, Ju Kim, Yong-Suk Jang

**Affiliations:** 1Department of Bioactive Material Sciences and Research Center of Bioactive Materials, Jeonbuk National University, Jeonju 54896, Republic of Korea; byeolheecho@jbnu.ac.kr; 2Department of Molecular Biology and the Institute for Molecular Biology and Genetics, Jeonbuk National University, Jeonju 54896, Republic of Korea; ju226@jbnu.ac.kr

**Keywords:** adjuvant, recombinant antigen, papain-like protease, SARS-CoV-2, vaccine

## Abstract

Most of the licensed vaccines against SARS-CoV-2 target spike proteins to induce viral neutralizing antibodies. However, currently prevalent SARS-CoV-2 variants contain many mutations, especially in their spike proteins. The development of vaccine antigens with conserved sequences that cross-react with variants of SARS-CoV-2 is needed to effectively defend against SARS-CoV-2 infection. Given that viral infection is initiated in the respiratory mucosa, strengthening the mucosal immune response would provide effective protection. We constructed a mucosal vaccine antigen using the papain-like protease (PLpro) domain of non-structural protein 3 of SARS-CoV-2. To potentiate the mucosal immune response, PLpro was combined with human beta-defensin 2, an antimicrobial peptide with mucosal immune adjuvant activity, and Co1, an M-cell-targeting ligand. Intranasal administration of the recombinant PLpro antigen conjugate into C57BL/6 and hACE2 knock-in (KI) mice induced antigen-specific T-cell and antibody responses with complement-dependent cytotoxic activity. Viral challenge experiments using the Wuhan and Delta strains of SARS-CoV-2 provided further evidence that immunized hACE2 KI mice were protected against viral challenge infections. Our study shows that PLpro is a useful candidate vaccine antigen against SARS-CoV-2 infection and that the inclusion of human beta-defensin 2 and Co1 in the recombinant construct may enhance the efficacy of the vaccine.

## 1. Introduction

The rapid spread of severe acute respiratory syndrome coronavirus 2 (SARS-CoV-2) fueled the development of vaccines to reduce the morbidity and mortality associated with coronavirus disease (COVID-19) such that various vaccines are currently approved and used worldwide [1]. However, the continuous replication of SARS-CoV-2 in animals and humans risks the emergence of variants able to cause zoonotic infections readily transmissible among humans. Therefore, the risk of another coronavirus pandemic outbreak remains. In addition, because current vaccines do not completely prevent re-infection, alternative coronavirus vaccine designs providing broad protective efficacy are needed. Currently available vaccines stimulate the production of neutralizing antibodies (Abs) against viral surface proteins, particularly the receptor-binding domain (RBD), thereby preventing the virus from the receptor binding required for viral entry and in turn infection. Accordingly, spike (S) protein is a common target for coronavirus vaccine research, and SARS-CoV-2 S protein is also the target of available COVID-19 vaccines [2,3,4]. However, variants of SARS-CoV-2 have mutations in S proteins, including in the RBD region, which can reduce the efficacy of these vaccines [5]. For example, the BNT162b2 vaccine which targets the Wuhan strain of SARS-CoV-2 induced 1.7- to 2.4-fold reduced neutralizing Ab efficacy against a variant with the D614G mutation [6]. Sera from individuals who received two doses of the BNT162b2 or mRNA-1273 vaccine showed increased neutralization of variants compared to those who received one dose. However, the neutralization efficiency against P.1 (Gamma), which has a high mutation rate in the S protein, decreased 6.7- and 4.5-fold for BNT162b2 and mRNA-1273, respectively, compared to wild-type virus, and decreased 34.5- and 27.7-fold against B.1.351 v1 (Beta) for BNT162b2 and mRNA-1273, respectively [7]. In addition, serum from COVID-19 patients has significantly decreased neutralization efficiency for the B.1.351 variant compared to the Wuhan and D614G variants [8]. These results suggest the need for studies on vaccines that are effective against the new variants.

SARS-CoV-2 is an enveloped, single-stranded, positive-sense RNA virus belonging to the *Betacoronavirus* genus of the *Coronaviridae* family [9,10]. It possesses two cysteine proteases: papain-like protease (PLpro), encoded by non-structural protein 3 (Nsp3), and 3C-like protease (3CLpro), embedded in Nsp5; both are required for the cleavage of polyproteins. PLpro is involved in the proteolytic maturation of viral polyproteins and immune modulation through the deubiquitinase and deISGylation activities of host proteins [11,12]. It is therefore considered a valid two-pronged antiviral target. Epitope mapping studies demonstrated the potential of 3CLpro and PLpro as well as viral M, N, and ORF3 proteins as novel vaccine target proteins, based on the presence of T-cell and B-cell epitopes [13,14,15,16,17]. Ong E et al. reported that several MHCI and MHCII epitopes, but not B-cell epitopes, were found in PLpro [17]. However, epitope profiling in individuals with COVID-19 and healthy pre-pandemic individuals showed that common B-cell epitopes of human coronaviruses were present in PLpro in orf1ab [16]. Additionally, variants of SARS-CoV-2, including Alpha, Beta, Gamma, Delta, and Omicron, have contained fewer mutations in PLpro than in the S protein [18]. These observations indicate that PLpro is highly conserved among coronaviruses and thus is a potential cross-protective antigen (Ag) among variants [19].

The mucosal immune system serves as the primary defense against pathogens, including bacteria, viruses, and dietary Ags. Within this system, nasal-associated lymphoid tissue (NALT) is an inductive site for the mucosal immune response against inhaled pathogens [20]. NALT is covered with epithelial cells that contain microfold (M) cells, which are specialized epithelial cells located on the follicle-associated epithelia overlying NALT [21]. M-cells play a crucial role in efficiently inducing mucosal immune responses against inhaled pathogens, by transporting Ags to NALT via transcytosis [22]. Mucosal immune responses elicited in NALT can also result in systemic immune responses [23]. Therefore, the induction of a mucosal immune response is considered an effective defense mechanism for pathogens that infect through respiratory mucosa, such as SARS-CoV-2. However, despite the advantages of mucosal vaccines, the low efficiency of Ag delivery through mucous membranes must be overcome to induce effective mucosal immune responses. Previously, we explored a Co1 peptide as an M-cell-targeting ligand capable of eliciting a mucosal immune response; the peptide was identified by bio-panning a phage display library against an in vitro M-cell co-culture system [24]. The adjuvant function of Co1 ligand was demonstrated by its binding to the C5a receptor (C5aR) expressed on M-cells in Peyer’s patches, which in turn enhanced Ag-specific humoral and cytotoxic T lymphocyte (CTL) responses [24,25,26]. Additionally, the Co1 ligand was shown to stimulate chemotaxis and Ag cross-presentation, leading to an increase in CTL responses through its interaction with C5aR^+^ lysogenic dendritic cells (LysoDCs) [27,28]. Co1 thus seems to act as an adjuvant to enhance the Ag-specific immune response through the mucosal immune system.

Host defense peptides play important roles in systemic and mucosal tissues by providing a primary defense system that protects the host from various pathogens and maintains tissue homeostasis [29,30]. They include human beta-defensins (HBDs), a group of antimicrobial peptides expressed in the skin and mucosal epithelium. HBD2 exhibits antimicrobial activity against Gram-negative bacteria and fungi [31,32], in addition to inhibiting the replication of HIV by increasing the expression of the antiretroviral factors APOBEC3G and APOBEC3A in human monocyte-derived macrophages [33]. Other than its antimicrobial functions, HBD2 establishes a connection between the induction of pathogen-specific innate and adaptive immunity and the activation of leukocyte subsets, such as macrophages, DCs, and T-cells [34,35,36]. We previously reported that HBD2 binds to CCR2, a chemokine receptor, in THP-1 cells, thus increasing the levels of chemokines and cytokines involved in the antiviral immune response, including IFN-β, IFN-γ, MxA, PKR, RNaseL, and NOD2. Ultimately, HBD2 is thought to promote innate immunity and thereby enhance the adaptive immune response [37,38]. The fact that the intranasal administration of viral Ag conjugated with HBD2 elicits stronger protective immune responses than immunization via intramuscular injection [39] suggests that HBD2 can be utilized as an adjuvant to induce local and systemic immune responses through the mucosal immune system.

In this study, we explored the potential of PLpro as a vaccine target Ag. In addition, by fusing the two immunoadjuvants described above, Co1 and HBD2, we were able to produce recombinant PLpro Ag, which improved the mucosal immunogenicity of PLpro. Intranasal administration of recombinant PLpro Ag effectively induced both protective immune responses to SARS-CoV-2 and systemic and mucosal immunity, including Ag-specific Ab and T-cell responses.

## 2. Materials and Methods

### 2.1. Materials and Experimental Animals

Seven-week-old C57BL/6 female mice (Koatech Laboratory Animal Center, Pyeongtaek, Republic of Korea) and hACE2-All CDS B6J knock-in (hACE2 KI) mice (Cyagen, Santa Clara, CA, USA) were housed under specific pathogen-free (SPF) conditions with food and water available ad libitum. The animal experiments were conducted with the approval of the Institutional Animal Care and Use Committee of Jeonbuk National University (Approval no. JBNU 2022-081 and JBNU 2022-082) and according to the guidelines of the Committee. SARS-CoV-2 Wuhan and Delta strains were acquired from the National Culture Collection for Pathogens (NCCP) of Korea (Cheongju, Republic of Korea). Experiments using SARS-CoV-2 were carried out in a biosafety level 3 facility of the Korea Zoonosis Research Institute (Iksan, Republic of Korea) of Jeonbuk National University, following the World Health Organization’s recommendations. Chemicals and laboratory products from SPL Life Sciences (SpL Life Sciences, Pocheon, Republic of Korea) and Sigma Chemical Co. (St. Louis, MO, USA) were used, respectively, unless otherwise noted.

### 2.2. Cloning and Expression of Proteins

PLpro (1564–1882 amino acids) of the Nsp3 of SARS-CoV-2 (Wuhan strain, also referred to as 2019-nCoV) was previously identified (reference sequence: YP_009725299.1). The gene for recombinant PLpro conjugated with HBD2 and Co1 was synthesized by Bioneer Inc. (Daejeon, Republic of Korea). This recombinant PLpro conjugate contained human beta-defensin 2 (sequence ID: AF071216.1) at its N-terminus and Co1 (SFHQLPARSPLP) at its C-terminus (Figure 1A). The PLpro gene (PLpro19) was amplified from the PLpro gene construct involving HBD2 and Co1 using the followed forward and reverse primer: 5′-CAT ATG GAG CTC ATG GAA GTG AGG ACT ATT AAG-3′ (underlining indicates the SacI restriction site) and 5′-GCT CTA GAT TAA TAA GTA ACT GGT TTT AT-3′ (underlining indicates the XbaI restriction site). The recombinant PLpro conjugate (HBD2-PLpro19-Co1) and PLpro (PLpro19) genes were cloned into pCold II (Takara Bio, Shiga, Japan) with a His-tag at the N-terminus. The pCold II expression constructs were transformed into BL21 star (DE3) competent Escherichia coli cells. Expression of recombinant Ags in the transformants was induced by adding IPTG (1 mM, Duchefa, Haarlem, The Netherlands). After removing soluble fractions by homogenizing the E. coli pellet, the pellet was washed five times with PBS. Inclusion bodies of PLpro19 and HBD2-PLpro19-Co1 were resolved with 8 M urea buffer. Proteins were dialyzed continuously with urea buffers (2 M, 1 M, 0.5 M) containing 400 mM arginine and three times with PBS. Recombinant Ags were confirmed by SDS-PAGE and Western blotting using anti-5× His tag (Qiagen, Hilden, Germany) and anti-PLpro (LifeSensors, Malvern, PA, USA) Abs.

### 2.3. Immunization and Challenge of Mice

C57BL/6 or hACE2 KI mice were immunized intranasally with 10 μg PLpro19 or HBD2-PLpro19-Co1 Ag once a week for 5 weeks, and PBS was used as the negative control. To evaluate the Ag-specific Ab response, serum was obtained 3 days after the last immunization, and bronchoalveolar lavage fluid (BALF) was obtained 7 days after. The spleens, NALTs, and lungs were also collected 7 days after the last immunization and assessed for Ag-specific T-cell responses. To assess the effect of Ag re-exposure, immunized mice were boosted via the intranasal route with each recombinant Ag 2 months after the last immunization. Samples and tissues were prepared as described previously [40].

SARS-CoV-2 was propagated in Vero E6 cells and used to assess morbidity and mortality induced in hACE2 KI mice. Immunized hACE2 KI mice were inoculated intranasally with 2 × 10^5^ plaque-forming units (PFUs) (Wuhan strain, NCCP 43326, S clade) or 1 × 10^5^ PFUs (Delta strain, NCCP 43405, GK clade) of SARS-CoV-2 after the fifth immunization. For up to 8 days after infection, the survival, weight, and pathological signs of the infected mice were observed. On day 8, tissue specimens were obtained from infected mice. Viral gene expression was measured by quantitative real-time reverse transcription–polymerase chain reaction (qRT-PCR).

### 2.4. Enzyme-Linked Immunosorbent Assay (ELISA)

To determine the levels of PLpro19-specific immunoglobulin G (IgG) and IgA in serum and BALF, PLpro19 protein (200 ng/well) was prepared in coating buffer and coated onto 96-well ELISA plates (Corning, Steuben County, NY, USA) overnight at 4 °C. After blocking, serial dilutions of serum or BALF were added to each well, followed by an overnight incubation at 4 °C. Next, alkaline phosphate-conjugated secondary Abs were added. Colors were developed by adding the appropriate substrate and quantified using SPECTROstar Nano (BMG LABTECH, Ortenberg, Germany). The amounts of Abs in sera and BALF were quantified from the ELISA results using standard curves, which were prepared by coating serially diluted normal mouse IgG or IgA on each well of an ELISA plate.

### 2.5. Complement-Dependent Cytotoxicity Assay

Vero E6 cells were obtained from the Korean Cell Line Bank (KCLB, Seoul, Republic of Korea) and incubated in a 96-well plate with SARS-CoV-2 (MOI: 0.1) for 24 h. Serum from immunized mice was diluted 100-fold with cell culture medium and then mixed with normal mouse serum at a ratio of 1:2. The mixture was added to Vero E6 cells after removal of the medium. After a 4 h incubation, the cells were detached with a PBS-EDTA solution and fixed with 4% paraformaldehyde at 4 °C for 20 min. Following a washing step, they were incubated with anti-C3 Ab (Invitrogen, Waltham, MA, USA), followed by anti-goat IgG-Alexa Fluor 647 (Invitrogen). The cells were permeabilized for 30 min with perm/wash buffer (BD Bioscience, Franklin Lakes, NJ, USA) and incubated first with anti-SARS-CoV-2 RBD Ab (Invitrogen) and then with anti-rabbit IgG-Alexa Fluor 555 (Invitrogen). They were then analyzed using a flow cytometer (CytoFLEX; Beckman Coulter, Brea, CA, USA).

### 2.6. Flow Cytometry

The spleens, NALTs, and lungs of the immunized mice were collected and cells were obtained by digesting the chopped tissue with RPMI-1640 containing 10% FBS, DNase I, and collagenase (Roche, Basel, Switzerland). Lymphocytes were separated by Percoll (Cytiva, Marlborough, MA, USA) density gradient centrifugation and then incubated with 1 μg of PLpro19 for 24 h at 37 °C in a CO_2_ incubator. For granzyme B (Grz B) staining, GolgiPlug^TM^ containing brefeldin A and GolgiStop^TM^ containing monensin were added, followed by an incubation of at least 5 h. After Fc receptor blocking with anti-mouse CD16/32 Ab (Invitrogen), the cells were stained with the indicated fluorochrome-conjugated Abs in staining solution: anti-mouse CD3-APC-Cy7 (BioLegend, San Diego, CA, USA), anti-mouse CD4-PerCP-vio700 (Miltenyi Biotec, Bergisch Gladbach, Germany), anti-mouse CD8-PE (Miltenyi Biotec), and anti-mouse CD44-PE-Cy770 (Miltenyi Biotec). For intracellular staining, anti-mouse Grz B-PE-Cy7 (Biolegend) was used in cells permeabilized using the BD Cytofix/Cytoperm™ fixation/permeabilization kit (BD Bioscience). The samples were stained as described above and subjected to a flow cytometric analysis (CytoFLEX; Beckman Coulter); CytExpert 2.4 software (Beckman Coulter) was used to analyze the data. Cytokines in the culture supernatants of Ag-stimulated T-cells were quantified using a cytometric bead array mouse Th1/Th2/Th17 cytokine kit (BD Bioscience), following the guidelines provided by the manufacturer. Data analysis was conducted using FCAP Array software V3.0.1 (BD Bioscience).

### 2.7. Quantitative RT-PCR Analysis

Lung samples from SARS-CoV-2-infected hACE2 KI mice were transferred to individual vials containing TRIzol reagent (Thermo Fisher Scientific, Waltham, MA, USA) and stored at –80 °C. Collected lung tissues were ground and subjected to total RNA isolation using a total RNA prep kit (Biofact, Daejeon, Republic of Korea). The extracted RNA was converted to cDNA using a Reverse-Transcription Kit (Promega, Madison, WI, USA). qRT-PCR for gene expression analysis was performed using a CFX Connect Real-Time System (Bio-Rad Laboratories, Hercules, CA, USA) with the SsoAdvanced Universal SYBR Green Supermix (Bio-Rad). Initial denaturation of the template was carried out at 95 °C for 3 min, followed by 40 cycles of denaturation at 95 °C for 15 s, annealing at 52 °C for 30 s, and extension at 72 °C for 30 s. The primer sequences were as follows: S protein of SARS-CoV-2, forward 5′-CCY ACM AAG CTG AAY GAC CTG TGC TTY ACM-3′ and reverse 5′-CAG GCR ATC ACR CAG CCG GTG-3′; N protein of SARS-CoV-2, forward 5′-ATG CTG CAA TCG TGC TAC AA-3′ and reverse 5′-GAC TGC CGC CTC TC-3′ [19]; mouse β-actin, forward 5′-CGT ACC ACA GGC ATT GTG A-3′ and reverse 5′-CTC GTT GCC AAT AGT GAT GA-3′ [36]. The relative gene expression level was normalized to the endogenous control gene (mouse β-actin); fold-change calculations were performed using CFX Maestro software Version 4.0 (Bio-Rad Laboratories).

### 2.8. Statistical Analysis

Statistical analysis consisted of one-way and two-way ANOVAs. The results are presented as the standard error of the mean (SEM) or standard deviation (SD). The analyses were performed using GraphPad Prism 8 (GraphPad Software Inc., La Jolla, CA, USA).

## 3. Results

### 3.1. Intranasal Administration of Recombinant PLpro Conjugate Elicits Mucosal and Systemic Immune Responses in C57BL/6 Mice

PLpro, also known as the coronavirus multifunctional Nsp3 domain, encompasses amino acids 1564–1882 of the orf1ab polyprotein of SARS-CoV-2. We constructed a recombinant PLpro by conjugating HBD2 and Co1 to the N- and C-termini, respectively, of PLpro (Figure 1A). The recombinant protein was expressed in *E. coli*, and soluble forms were purified and used for immunization. Ags were purified to 90–95% purity and PLpro19 and HBD2-PLpro19-Co1 were obtained at 0.5 and 0.13 mg/L, respectively. The ability of recombinant PLpro conjugated with HBD2 and Co1 to enhance immunogenicity was determined by intranasally immunizing C57BL/6 mice with PLpro19 or HBD2-PLpro19-Co1 and then measuring the levels of Ag-specific Abs in serum and BALF (Figure 1B). Higher levels of PLpro19-specific IgG were detected in the sera of mice immunized with HBD2-PLpro19-Co1 than with PLpro19 alone (200.28 ± 73.94 vs. 62.77 ± 33.93 μg/mL), but the level of PLpro19-specific serum IgA did not significantly differ between the two groups. Importantly, the level of PLpro19-specific IgG (*p* < 0.05) and IgA (*p* < 0.01) in BALF was significantly higher in mice immunized with HBD2-PLpro19-Co1 than with PLpro19 alone. These results suggest that the recombinant PLpro construct containing HBD2 and Co1 is capable of inducing Ag-specific mucosal and systemic immune responses.

Because PLpro is not a structural protein mainly exposed on the surface of the virus, PLpro-specific Abs might not be effective in neutralizing or capable of directly eliminating viruses. The complement-dependent cytotoxicity (CDC) of the Abs in serum and BALF was therefore examined. A previous study showed that low levels of ORF1ab proteins are expressed on the surface of SARS-CoV-2-infected cells [13]. Similarly, we found that PLpro19-specific sera recognize and bind to SARS-CoV-2-infected Vero E6 cells (Appendix A). The recognition and binding of virus-infected cells was enhanced by the use of sera from mice immunized with HBD2-PLpro19-Co1. Sera from PLpro19-immunized mice were also able to bind to infected cells, thus demonstrating that PLpro19-specific Abs can bind to the surface of SARS-CoV-2-infected cells. Normal mouse serum was then added as a complement source in the CDC assay, and SARS-CoV-2-infected VeroE6 cells were cultured with sera and BALF from the immunized mice. Treatment with serum from mice immunized with PLpro19 or HBD2-PLpro19-Co1 significantly increased the proportion of SARS-CoV-2 RBD^+^ and complement 3 (C3)^+^ cells (Figure 1C). In the CDC assay using BALF in the group immunized with PLpro19 or HBD2-PLpro19-Co1, there was a slight, although nonsignificant, increase in SARS-CoV-2 RBD^+^ and C3^+^ cells. The main subclasses of PLpro19-specific IgG induced by immunization were IgG1 and IgG2b (Figure 1D). The main subclasses of IgG that mediate CDC and ADCC in mice are IgG2a and IgG2b [41]. In addition, IgG1 has been reported to have CDC potential, and high levels of IgG1 in immunized mice promote the removal of virus-infected cells [42]. These results indicated (1) that anti-PLpro19 Abs can bind to SARS-CoV-2-infected cells to activate the complement system, leading to the lysis of infected cells and therefore that (2) intranasal administration of HBD2-PLpro19-Co1 stimulates the production of CDC-inducing PLpro19 Abs to a greater extent than PLpro19 alone, thereby enhancing protection against SARS-CoV-2 infection and transmission.

### 3.2. Recombinant PLpro Conjugate Stimulates Ag-Specific CTL Activation and the Production of Cytokines Involved in T-Cell Immune Responses

The Co1 peptide induces an Ag-specific CTL response [26]. In this study, PLpro19-specific CTL responses were induced in mice 7 days after the last immunization with PLpro19 or HBD2-PLpro19-Co1. In particular, the proportion of Grz B-producing CD8^+^ T-cells in the lungs was significantly higher in mice immunized with HBD2-PLpro19-Co1 than in those immunized with PLpro alone (12.09% ± 2.01% vs. 4.41% ± 0.72%, *p* < 0.05) (Figure 2A). An increased proportion of Grz B^+^ CD8^+^ T-cells in the spleens of mice immunized with HBD2-PLpro19-Co1 was also observed. An analysis of cytokine production from lymphocytes prepared from spleen, lung, and NALTs exposed to PLpro19 for 24 h showed a significant increase in the expression levels of IFN-γ, TNF-α, IL-2, IL-17A, and IL-6 in mice immunized with HBD2-PLpro19-Co1 (Figure 2B). These results showed that intranasal administration of HBD2-PLpro19-Co1 induces cytotoxic CD8^+^ T-cell responses as well as Th1, Th2, and Th17 immune responses. Thus, the conjugation of PLpro with HBD2 and Co1 induced both humoral and cellular immune responses locally and systemically after intranasal administration.

### 3.3. The Persistence of an Ag-Specific Adaptive Immune Response Induced by PLpro Conjugated with HBD2 and Co1 Allows for the Retention of Immune Memory

The induction of immune memory is necessary for long-term vaccination efficacy because it enables an effective and rapid activation of the desired immune responses after Ag re-exposure, resulting in protection against specific virus infection. To assess the ability of the recombinant PLpro conjugate to maintain Ag-specific immune responses, mice were boosted 2 months after the last immunization, and cellular immune responses, including CTL activity and Ab responses specific to PLpro, were then measured. Mice immunized with HBD2-PLpro19-Co1 maintained higher levels of Ag-specific IgG and IgA in both serum and BALF (Figure 3A). Additionally, the proportion of Grz B^+^ CD8^+^ T-cells was increased in the spleen and lungs of the mice immunized with PLpro19 or HBD2-PLpro19-Co1 compared to those with PBS (Figure 3B). In particular, a significantly higher proportion of Grz B^+^ CD8^+^ T-cells was found in the lung lymphocytes of HBD2-PLpro19-Co1-immunized mice than in those of the PBS group (3.10% ± 0.06% vs. 1.80% ± 0.19%, *p* < 0.05). Th1, Th2, and Th17 cytokines were also highly expressed in lymphocytes prepared from the spleen, lungs, and NALTs of the HBD2-PLpro19-Co1-immunized group (Figure 3C). In lymphocytes isolated from the lungs, the proportion of memory CD4^+^ and CD8^+^ T-cells highly expressing CD44 was significantly larger in HBD2-PLpro19-Co1-immunized mice than in PLpro19-immunized mice (47.69% ± 1.32% vs. 33.88% ± 0.98%, *p* < 0.01, and 21.91% ± 1.60% vs. 13.58% ± 0.47%, *p* < 0.05, respectively) (Figure 3D). These results suggest that the conjugation of PLpro with HBD2 and Co1 can induce efficient memory T-cell responses at sites of initial virus entry and replication, resulting in humoral and cellular immune responses against mucosal and systemic viral infection.

### 3.4. Intranasal Administration of Recombinant PLpro Conjugate Induces Protective Immune Responses against SARS-CoV-2 in hACE2 KI Mice

The ability of the immune response induced by HBD2-PLpro19-Co1 to effectively protect against SARS-CoV-2 infection was assessed in a virus challenge experiment using hACE2 KI mice, whose tissues have a high binding affinity for SARS-CoV-2. The mice were intranasally immunized with a recombinant PLpro conjugate according to the same schedule applied in the C57BL/6 mice, and the induction of Ag-specific Ab responses was analyzed in serum and BALF samples. The results showed that immunization of HBD2-PLpro19-Co1 induced Ag-specific Ab and T-cell responses in hACE2 KI mice (Figure 4), including a significant increase in the level of PLpro19-specific serum IgG and IgA compared to the PBS group (3469.79 ± 1059.25 vs. 29.60 ± 3.25 μg/mL, *p* < 0.05 and 0.30 ± 0.03 vs. 0.00 ± 0.00 μg/mL, *p* < 0.0001) (Figure 4A). PLpro19-specific IgG and IgA levels in BALF were also higher in the HBD2-PLpro19-Co1-immunized group than in the PBS group (Figure 4A). Lymphocytes were isolated from the spleens and lungs 7 days after the last immunization to assess the activation of Ag-specific CTL immune responses. The proportion of Grz B^+^ CD8^+^ T-cells in the spleens and lungs was significantly higher in hACE2 KI mice immunized with HBD2-PLpro19-Co1 than in the PBS group (3.89% ± 0.33% vs. 0.59% ± 0.04%, *p* < 0.01, and 3.99% ± 0.37% vs. 1.61% ± 0.21%, *p* < 0.05) (Figure 4B). In particular, the expression levels of IFN-γ and IL-6 were significantly higher in both mucosal and systemic lymphoid tissues of hACE2 KI mice immunized with HBD2-PLpro19-Co1 (Figure 4C). These findings suggest that immunization with HBD2-PLpro19-Co1 results in a robust cytotoxic CD8^+^ T-cell response and cellular immune responses in hACE2 KI mice.

Next, the ability of the immune responses induced by HBD2-PLpro19-Co1 to protect against SARS-CoV-2 challenge infection was determined by intranasally immunizing hACE2 KI mice with recombinant PLpro proteins five times once a week, followed by an intranasal inoculation with 2 × 10^5^ PFUs of SARS-CoV-2 (Wuhan strain) (Figure 5A,B). Eight days after SARS-CoV-2 challenge infection, all of the mice were still alive, without any disease symptoms. However, transcript levels of the S gene of SARS-CoV-2 in the lungs were significantly lower in the HBD2-PLpro19-Co1-immunized mice than in the PBS-immunized mice (*p* < 0.05). A challenge experiment was then conducted in hACE2 KI mice immunized with HBD2-PLpro19-Co1, PLpro19, or PBS by inoculating them with 1 × 10^5^ PFUs of the highly infectious SARS-CoV-2 Delta variant. Survival, weight, and pathological manifestations were monitored for 8 days (Figure 5C,D). While there was no major difference in body weight between the two groups of mice, the transcript level of the N gene of SARS-CoV-2 in the lungs was significantly lower in the HBD2-PLpro19-Co1-immunized group than in the PLpro19- or PBS-immunized mice (*p* < 0.05, *p* < 0.01, respectively) (Figure 5D). Intranasal immunization of HBD2-PLpro19-Co1 therefore inhibited SARS-CoV-2 infection after challenge infection in hACE2 KI mice. Collectively, these results demonstrated the ability of HBD2-PLpro19-Co1 to protect the mice against both the Wuhan strain and Delta variant of SARS-CoV-2 infection by inducing humoral and CTL responses.

### 3.5. Long-Term Persistence of Ag-Specific Adaptive Immune Response Is Enhanced by PLpro Conjugated with HBD2 and Co1 in hACE2 KI Mice

The Ag-specific immune responses were maintained for an extended period in hACE2 KI mice. Similar to C57BL/6 mice, hACE2 KI mice immunized with HBD2-PLpro19-Co1 exhibited higher levels of PLpro19-specific Ab and T-cell immune responses 2 months after the last immunization (Figure 6). For example, the levels of PLpro19-specific IgG and IgA were higher in the sera and BALFs of the HBD2-PLpro19-Co1-immunized mice than in the PBS group (Figure 6A). HBD2-PLpro19-Co1-immunized mice also had a significantly higher proportion of lung and spleen cytotoxic CD8^+^ T-cells expressing Grz B (7.11% ± 0.26% vs. 1.20% ± 0.14%, *p* < 0.0001, and 6.64% ± 0.14% vs. 2.89% ± 0.40%, *p* < 0.01) (Figure 6B). Furthermore, higher levels of CD4^+^ memory T-cells in lung lymphocytes were determined in mice immunized with PLpro19 or HBD2-PLpro19-Co1 than in PBS-immunized mice (49.33% ± 1.81% vs. 23.00% ± 0.81%, *p* < 0.0001, and 58.02% ± 1.09% vs. 23.00% ± 0.81%, *p* < 0.0001) (Figure 6C). However, there were no significant differences in CD8^+^ memory T-cells among the different groups. These results showed that conjugation of HBD2 and Co1 with the viral PLpro Ag can maintain Ag-specific immune memory at the site of initial viral infection and induce both mucosal and systemic protective immune responses.

## 4. Discussion

SARS-CoV-2 is responsible for COVID-19, which was first reported in December 2019. The World Health Organization categorized COVID-19 as a pandemic on 11 March 2020. The urgency of the situation fueled the rapid development of COVID-19 vaccines through global cooperation among researchers employing strategies quite different from traditional approaches to vaccine development [43]. However, variants of SARS-CoV-2 continue to emerge [44], with many mutations occurring in the S protein, a main target of neutralizing Abs in coronavirus infections. Neutralizing Abs from early SARS-CoV-2 vaccine recipients are unable to neutralize other variants [14]. As a result, many of the mutations in the spike region involved in host receptor binding could interfere with the efficacy of existing vaccines. This has motivated efforts to develop a vaccine with appropriate efficacy against the many different variants of SARS-CoV-2 [45]. To overcome antigenic variation, conserved sequences have been used as vaccine Ags against various types of influenza virus. The hemagglutinin (HA) stem domain, matrix protein 1 (M1), and nucleoprotein (NP), which have high sequence homology among influenza viruses, are being applied as Ags and are being researched in clinical trials [46]. This strategy was also applied to vaccines against SARS-CoV-2, such that heptad repeat 1 (HR1) in the S2 region of SARS-CoV-2, a sequence conserved among variants, was used as a vaccine Ag and tested in rabbits, mice, hamsters, and rhesus macaques [47]. Neutralizing Abs were induced by immunization with HR1-containing Ag and the immunized animals were protected against variant viruses. In addition, a mixture of RBD proteins containing five peptides, and conserved regions of the membrane (M), nucleocapsid (N), and S2 proteins of the variant, was proposed as a candidate pan-SARS-CoV-2 booster vaccine [48] and induced a T-cell response against the Delta variant and neutralized Abs.

Enhancing mucosal immunity against viruses infecting the respiratory mucosa, such as SARS-CoV-2 and influenza viruses, is likely to be the most effective strategy for preventing virus transmission and infection [15,16]. The available SARS-CoV-2 vaccines induce a systemic immune response via intramuscular injection. Secretory IgA (SIgA) against SARS-CoV-2 and influenza virus are induced by intramuscular vaccination, but with low Ab titers [49,50]. Mucosal vaccines can induce high-level SIgA expression in the mucosa [51]. Moreover, mucosal vaccines can protect against respiratory viruses because they induce immune responses not only in the mucosa (where viral infection begins) but also systematically [51]. In addition, mucosal vaccination increases the number of resident T-cells in tissues [52]. Consequently, we believe that intranasal vaccination could rapidly induce a strong defense response against respiratory viruses, especially at the respiratory mucosa. The mucosal vaccine platform presented in this study utilizes the PLpro of SARS-CoV-2 as an Ag and HBD2 and Co1 as Ag-targeting and immune-potentiating materials. Previous studies showed that Co1 and HBD2 enhance Ag delivery to the mucosal immune system and improve mucosal and systemic immune responses, respectively [26,39]. We showed that intranasal administration of the recombinant vaccine resulted in the production of PLpro19-specific Abs, representing both mucosal and systemic immunity and possessing CDC activity (Figure 1C). Virus-specific Abs, particularly virus-neutralizing Abs, play a crucial role in eliminating viruses and preventing infection by target viruses [17]. For example, Ab-mediated responses defend against target viruses through mechanisms such as Ab-dependent cellular cytotoxicity, Ab-dependent cellular phagocytosis, and CDC [18,53,54,55]. The NP of influenza is not an envelope protein and is therefore not exposed on the surface of the virus. However, it can be anchored and labeled on the surface of virus-infected host cells [56]. Abs that recognize cell-surface-exposed NP can bind to it, thus activating the complement system and inducing lysis or eliminating the infected cells by the cytotoxicity of natural killer cells with Fc receptors [57]. Additionally, virus-infected cells can be eliminated through Ab-mediated phagocytosis [58]. Therefore, Abs with these functions act as a defense mechanism against the virus. However, PLpro is not a structural protein exposed on the surface of SARS-CoV-2, and PLpro-specific Abs might not be effective in neutralizing or directly removing the virus. It has also been reported that the proportion of Nsp3 containing PLpro among the expressed viral proteins of SARS-CoV-2-infected cells is very low [13], so Abs produced by recombinant PLpro protein recognize and bind to infected cells (Appendix A). The function of PLpro-specific Abs was then confirmed in a CDC assay.

IgG is recognized by FcγR-expressing antigen-presenting cells and can induce ADCC or phagocytosis [59]. It can mediate complement activity, which may delay or inhibit virus propagation by removing virus-infected cells via CDC [59]. In mice, the main IgG isotypes that mediate CDC and ADCC are IgG2a and IgG2b. As shown in Figure 1D, the main isotypes of PLpro19-specific IgG induced by immunization were IgG1 and IgG2b. Given the possibility that IgG1 can mediate CDC [42], the high levels of IgG1 and IgG2b detected in immunized mice likely promoted the removal of virus-infected cells. In the case of mouse IgA, unlike humans, it has only one IgA isoform and does not have a functional homologue with FcαRI [60]. However, IgA-mediated effector functions via other IgA receptors including Fcα/μR (CD351) and transferrin R (CD71), whose expression has been reported in mice, are conceivable [59,61]. IgAs bind to PLpro proteins exposed on virus-infected cells in the mucosa and are recognized by FcR-expressing antigen-presenting cells, which may delay and interfere with virus transmission via endocytosis and/or phagocytosis [59]. We believe that neutralization through direct binding of SIgA is difficult because PLpro-specific SIgA cannot bind to viral particles. However, increased SIgA can induce immune exclusion, making it difficult to access the mucus membrane by capturing the pathogen in the mucus layer [62]. Therefore, it is thought that increased SIgA has the potential to reduce viral infection and transmission.

Co1 binds to C5aR on M cells, resulting in increased Ag delivery to the mucosal immune system and improved CTL responses [26]. Moreover, promotion of cross-presentation by the C5a/C5aR signaling pathway on C5aR^+^ LysoDCs resulted in an increase in CTL responses [27]. These findings are consistent with our observation of enhanced CTL responses following intranasal administration of HBD2-PLpro19-Co1. CD8^+^ CTLs express Grz B in the spleen and lungs, a major lymphoid tissue and an important target tissue of SARS-CoV-2 infection, respectively. In the early stages of viral infection, Grz B-expressing CD8^+^ T-cells specific to the target virus appear to be more effective in controlling the establishment of a viral reservoir than IFN-γ-expressing CD8^+^ T-cells [63]. Also, the proportion of Grz B-expressing CD8^+^ T-cells is higher in patients with mild symptoms than in those with severe symptoms, and Grz B is more effective in defending against the early stage of virus infection [64]. Interestingly, Grz B-expressing CD4^+^ T-cells were induced in ACE2 KI mice by the intranasal administration of HBD2-PLpro19-Co1 (Appendix A). Cytotoxic Grz B-expressing CD4^+^ T-cells were shown to effectively lyse virus-infected cells [65,66]. These results suggested that the mucosal and systemic immune responses induced by HBD2-PLpro19-Co1 protect against SARS-CoV-2 infection.

Challenge experiments conducted using the Wuhan strain and Delta variants in hACE2 KI mice, as an animal model of SARS-CoV-2 infection (Figure 5), did not result in the death of any of the mice. In contrast to hACE2 KI mice, hACE2 is overexpressed in epithelial cells throughout the body of K18-hACE2 mice, due to regulation of the epithelial cell-specific human keratin 18 (K18) promoter, resulting in more severe SARS-CoV-2 infection [67]. Therefore, it is difficult to compare mortality in hACE2 KI mice vs. K18-hACE2 mice, but the higher expression of hACE2 in the lung and nasal turbinate may facilitate the identification of SARS-CoV-2-induced lung damage in hACE2 KI mice [2,68]. These experiments demonstrated the protective effect of intranasally administered HBD2-PLpro19-Co1 against SARS-CoV-2, based on the level of viral gene expression in the lungs of immunized mice after virus challenge infection, detected using qRT-PCR (Figure 5). HBD2-PLpro19-Co1 may therefore be a useful mucosal vaccine Ag against SARS-CoV-2 infection, with PLpro serving as a promising vaccination target Ag.

## 5. Conclusions

This study showed that the Ag construct conjugated with HBD2 and Co1 increased Ag delivery to the mucosal immune system and thus enhanced Ag-specific humoral and cell-mediated immune responses in both mucosal and systemic immune compartments. Specifically, intranasal immunization with an Ag conjugated with HBD2 and the Co1 ligand enhanced Ag-specific cytotoxic CD8^+^ and CD4^+^ T-cell responses, as well as, interestingly, the production of Abs capable of inducing CDC. Therefore, PLpro Ag can be considered a target Ag for vaccination against SARS-CoV-2 variants. The recombinant construct may therefore provide a mucosal vaccine platform, with PLpro serving as a vaccine target Ag capable of inducing both Ab and T-cell immune responses against SARS-CoV-2.

## Figures and Tables

**Figure 1 vaccines-12-00441-f001:**
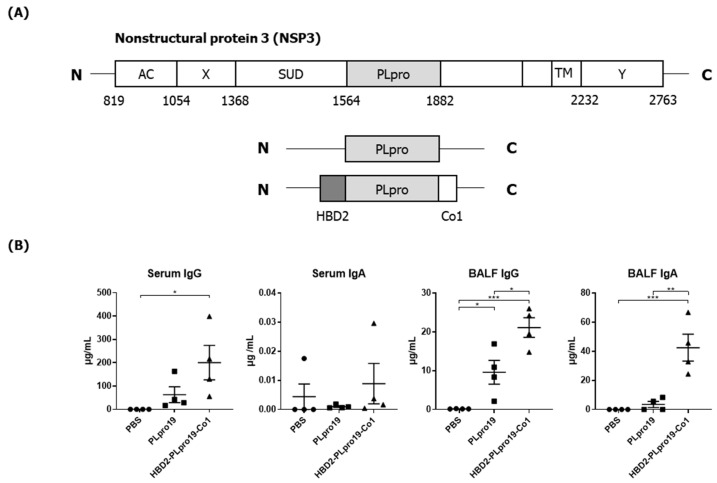
Ag-specific Ab responses in C57BL/6 mice immunized with recombinant proteins. (**A**) Recombinant protein containing the sequence of the PLpro domain in non-structural protein 3 (Nsp3) of SARS-CoV-2. (**B**) Mice were immunized intranasally 5 times with 10 μg of PLpro19 or HBD2-PLpro19-Co1 once a week. Ag-specific IgG and IgA in the serum and BALF of immunized C57BL/6 were analyzed by ELISA 1 week after the last immunization. (**C**) After a 1 h incubation of infected Vero E6 with serum from mice, SARS-CoV-R^+^C3^+^ Vero E6 cells were detected by flow cytometry. (**D**) Subclasses of PLpro19-specific IgG were analyzed by ELISA. Data are presented as means ± standard errors (SEs) of repeated experiments and analyzed in a one-way ANOVA (* *p* < 0.05, ** *p* < 0.01, *** *p* < 0.001, **** *p* < 0.0001, *n* = 4). AC, acidic domain; X, X-domain; SUD, SARS-unique domain; PLpro, Papain-like protease; TM, transmembrane region; Y, Y-domain.

**Figure 2 vaccines-12-00441-f002:**
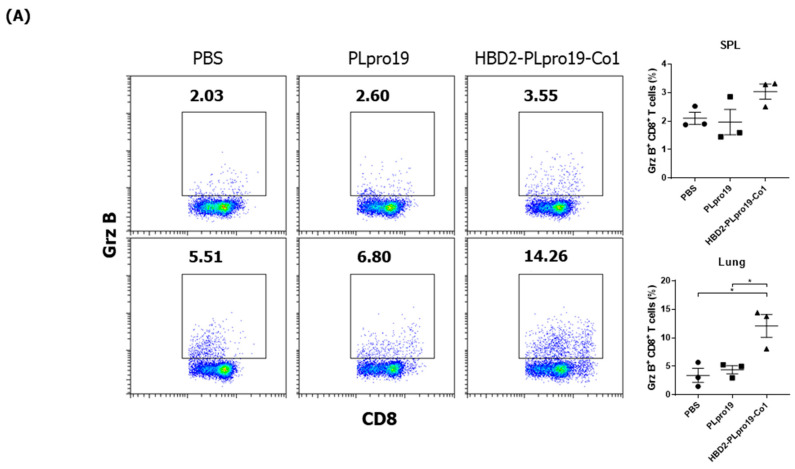
Ag-specific T-cell immune responses in C57BL/6 mice immunized with recombinant proteins. (**A**) Lymphocytes prepared from the lung and splenocytes were incubated with PLpro19 protein (1 μg) for 24 h. Grz B^+^ CD8^+^ cells were analyzed by flow cytometry. (**B**) Cytokines were detected by cytometric bead array (CBA). The experiments were repeated three times and representative results are shown. All data were analyzed in a one-way ANOVA (* *p* < 0.05, ** *p* < 0.01, *** *p* < 0.001, **** *p* < 0.0001, *n* = 3). The results are presented as means ± SE, with the exception of NALT (mean ± standard deviation [SD]).

**Figure 3 vaccines-12-00441-f003:**
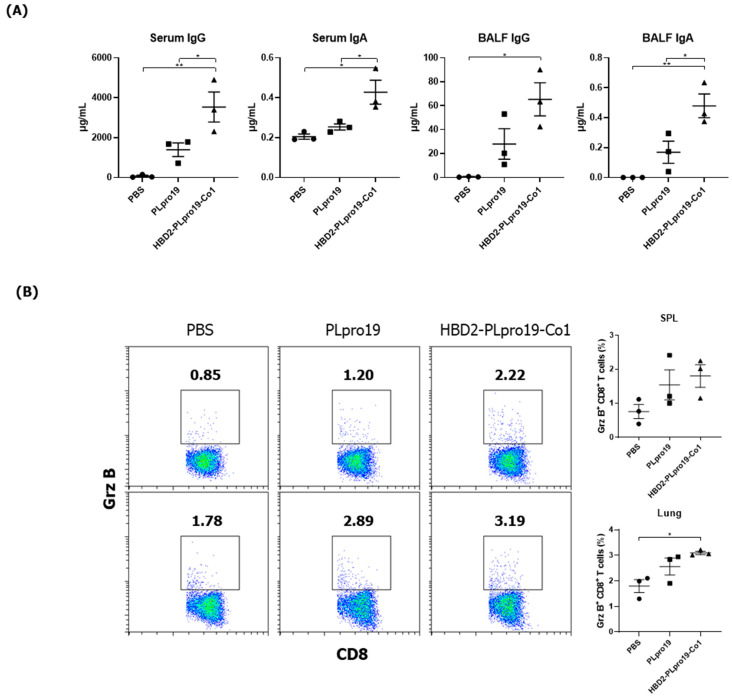
Maintenance of Ag-specific immune responses in mice. (**A**) C57BL/6 mice were boosted with 10 μg of PLpro19 or HBD2-PLpro19-Co1 2 months after the last immunization. PLpro19-specific IgG and IgA in the serum and BALF of immunized mice were detected by ELISA 1 week after boosting. (**B**) Lymphocytes from lung and splenocytes were incubated with the PLpro19 protein (1 μg) for 24 h. Grz B^+^ CD8^+^ cells were analyzed by flow cytometry. (**C**) Cytokines were detected by CBA. (**D**) CD44^hi^ CD4^+^ and CD8^+^ T-cells detected in the lung. The experiments were repeated twice and representative results are shown. The data were analyzed in a one-way ANOVA (* *p* < 0.05, ** *p* < 0.01, *** *p* < 0.001, **** *p* < 0.0001, *n* = 3) and are presented as means ± SE, with the exception of NALT (mean ± SD).

**Figure 4 vaccines-12-00441-f004:**
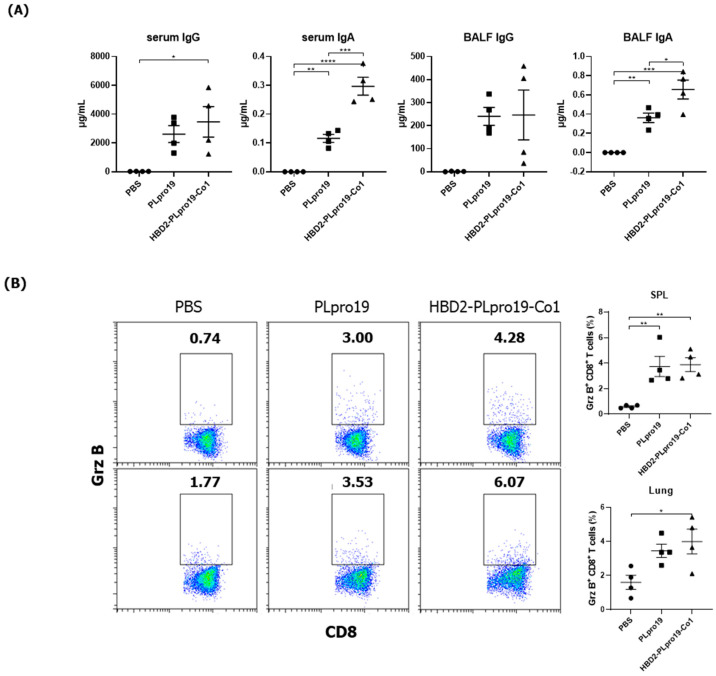
Immunogenicity of PLpro19 constructs in hACE2 KI mice. (**A**) hACE2 KI mice were immunized intranasally five times with 10 μg of PLpro19 or HBD2-PLpro19-Co1 once a week. The levels of PLpro19-specific IgG and IgA in the serum and BALF of the immunized hACE2 KI mice were determined by ELISA 1 week after the last immunization. (**B**) Cells isolated from the spleen and lung were stimulated with PLpro19 protein (1 μg) for 24 h. Grz B^+^ CD8^+^ T-cells were analyzed by flow cytometry. (**C**) Cytokines were detected by CBA. The experiments were repeated three times and representative results are shown. All data were analyzed in a one-way ANOVA (* *p* < 0.05, ** *p* < 0.01, *** *p* < 0.001, **** *p* < 0.0001, *n* = 4) and are presented as means ± SE, with the exception of NALT (mean ± SD).

**Figure 5 vaccines-12-00441-f005:**
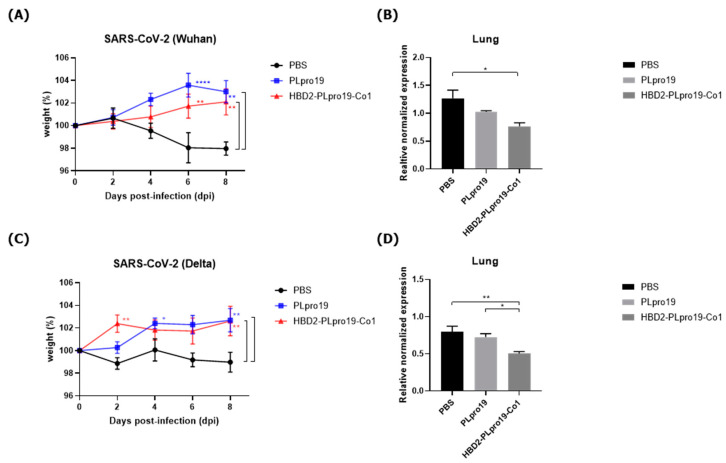
Protective immune response against SARS-CoV-2 challenge infection. (**A**) hACE2 KI mice (*n* = 7) were immunized five times with 10 μg of PLpro19 or HBD2-PLpro19-Co1 once a week. A week after the last immunization, the mice were infected intranasally with SARS-CoV-2. The mice were weighed daily for 8 days after infection with the Wuhan strain of SARS-CoV-2 (2 × 10^5^ PFUs). (**B**) RNA expression of the S protein gene of SARS-CoV-2 in lung was detected by qRT-PCR 8 days post-infection (*n* = 2). (**C**) hACE2 KI mice (*n* = 6 for PBS, *n* = 7 for PLpro19 and HBD2-PLpro19-Co1) were weighed daily for 8 days after infection with the Delta strain of SARS-CoV-2 (1 × 10^5^ PFUs). (**D**) RNA expression of the N protein of SARS-CoV-2 in the lung was detected by qRT-PCR 8 days after infection (*n* = 2). The experiments were repeated twice and representative results are shown. The changes in body weight of the mice were analyzed in a two-way ANOVA, and the qRT-PCR data in a one-way ANOVA (* *p* < 0.05, ** *p* < 0.01, **** *p* < 0.0001). The data are presented as means ± SE.

**Figure 6 vaccines-12-00441-f006:**
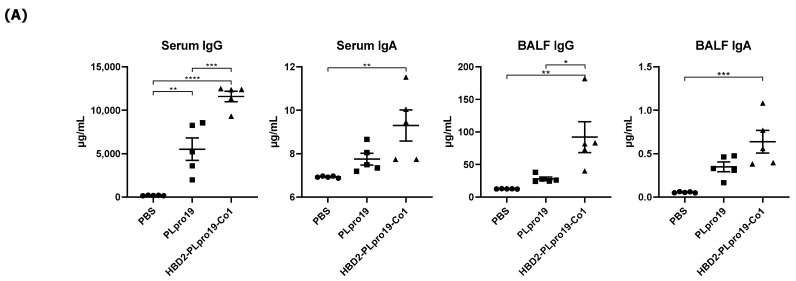
PLpro19-specific immune responses 2 months after the immunization of hACE2 KI mice. (**A**) hACE2 mice were boosted with 10 μg of PLpro19 or HBD2-PLpro19-Co1 2 months after the last immunization. PLpro19-specific IgG and IgA in the serum and BALF of immunized hACE2 KI mice were detected by ELISA 1 week after boosting. (**B**) Cells isolated from the spleen and lung of hACE2 KI mice were stimulated with PLpro19 protein (1 μg) for 24 h. Grz B^+^ CD8^+^ T-cells were analyzed by flow cytometry. (**C**) CD44^hi^ CD4^+^ and CD8^+^ T-cells were detected in the lung. The experiments were repeated three times and representative results are shown. All data were analyzed in a one-way ANOVA (* *p* < 0.05, ** *p* < 0.01, *** *p* < 0.001, **** *p* < 0.0001, *n* = 4). The data are presented as means ± SE.

## Data Availability

Data are contained within the article and Appendix A.

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
