# Peer review of "The Papain-like Protease Domain of Severe Acute Respiratory Syndrome Coronavirus 2 Conjugated with Human Beta-Defensin 2 and Co1 Induces Mucosal and Systemic Immune Responses against the Virus"

_vaccines, 2024, doi:10.3390/vaccines12040441_

Round 1

Reviewer 1 Report

Comments and Suggestions for Authors

This study aimed to determine the effectiveness of an adjuvanted M cell-targeting nasal vaccine (HBD-PLpro19-Co1) against SARS-CoV-2 virus infection. The authors showed that nasal immunization with HBD-PLpro19-Co1 induced increased levels of PLpro19-specific antibodies and cell-mediated immune responses in the lung and spleen, which provided protection against SARS-CoV-2 virus infection. In addition, the authors showed that nasal HBD-PLpro19-Co1 induced memory-type immune responses that were enhanced by booster immunization. Although the results are intriguing, the authors should address the following points to improve the quality of the manuscript.

Major Points:

1.    The authors did not provide the detailed method of the Ag-specific ELISA in the text or as a reference. Since the authors showed the quantitative amounts of antibodies (µg/ml), it is essential to show how these antibody titers are determined.

2.    It is important to show the effectiveness of an M cell-targeting nasal vaccine without HBD as a mucosal adjuvant.

3.    Since the authors suggested complement-dependent cytotoxicity (CDC), it is essential to determine PLpro19-specific IgG subclass responses.

4.    It is not clear that Figure 2A shows the cell population gated by CD8. Details should be provided. If Figure 2A represents CD8 gated cells, there are at least two or three subsets were recognized based on the fluorescence intensity (dull, medium, and very high). Since the staining pattern of CD8+ T cells is unusual (please compare with Figure 3A), the authors should provide an appropriate explanation.

5.    The cytokine beads array analysis was performed using T cell culture supernatants containing CD4+ T cells that can produce Th1, Th2, and Th17 cytokines in addition to CD8+ T cells. In this regard, the authors should specifically determine these cytokine responses by CD8+ T cells.

6.    The nasal virus challenge experiments should be performed 2 months after the last nasal immunization since hACE2 KI mice showed increased levels of antigen-specific IgG and IgA antibody responses in the lungs. 

7.    Since nasal immunization with HBD-PLpro19-Co1 induced significantly increased levels of antigen-specific IgG and IgA antibody responses in the respiratory tract, the authors should determine the role of each antibody isotype in the protection against viral challenge.

8.    The authors should discuss the strength of using the nasal HBD-PLpro19-Co1 vaccine since the induction of humoral and cell-mediated immunity is most likely achieved by systemic vaccination strategies.

Minor Point:

1.    C57BL6 mice should be written as C57BL/6 (Figure 1 legend).

Author Response

Reviewer #1

Comments and Suggestions for Authors

This study aimed to determine the effectiveness of an adjuvanted M cell-targeting nasal vaccine (HBD-PLpro19-Co1) against SARS-CoV-2 virus infection. The authors showed that nasal immunization with HBD-PLpro19-Co1 induced increased levels of PLpro19-specific antibodies and cell-mediated immune responses in the lung and spleen, which provided protection against SARS-CoV-2 virus infection. In addition, the authors showed that nasal HBD-PLpro19-Co1 induced memory-type immune responses that were enhanced by booster immunization. Although the results are intriguing, the authors should address the following points to improve the quality of the manuscript.

We thank Reviewer #1 for the positive evaluation of our manuscript and constructive suggestions to strengthen its quality. We have revised the manuscript to address the concerns. We hope that the revised manuscript has been sufficiently improved for acceptance for publication. Our responses to the specific points raised by the reviewer are provided below and indicated as grey background in the revised manuscript.

Major Points:

  1. The authors did not provide the detailed method of the Ag-specific ELISA in the text or as a reference. Since the authors showed the quantitative amounts of antibodies (µg/ml), it is essential to show how these antibody titers are determined.

             The amounts of antibodies contained in sera and BALF were quantified from the ELISA results using standard curves prepared by coating serially diluted normal mouse IgG and IgA onto each well of an ELISA plate. More details on quantification of antibodies have been added to the Materials and Methods section (page 4, lines 183-185).

  1. It is important to show the effectiveness of an M cell-targeting nasal vaccine without HBD as a mucosal adjuvant.

We reported the effectiveness of HBD2 and Co1 ligand conjugated with viral antigens as a mucosal vaccine adjuvant (Kim et al., 2018, Virol. J. 15:1; Kim et al., 2018, Cell. Immunol. 325:41). Based on these findings, we aimed to demonstrate the effectiveness of a construct containing both HBD2 and Co1 using PLpro antigen in this study.

  1. Since the authors suggested complement-dependent cytotoxicity (CDC), it is essential to determine PLpro19-specific IgG subclass responses.

             We agree and have analyzed the IgG subclasses of PLpro19-specific antibodies. In mice, the main IgG subclasses mediating CDC and ADCC are IgG2a and IgG2b (Gravina et al., 2023, Nat. Biotechnol. 41:717). The main IgG subclass capable of mediating CDC of PLpro19-specific IgG in this study was IgG2b. In addition, given that IgG1 has potential for CDC, we believe that the high level of IgG1 in immunized mice promotes the removal of virus-infected cells (Vukovic et al., 2023, Cancer Res. Commun. 3:109). We have added the above to lines 280-283 on page 6 and as Figure 1D, respectively.

  1. It is not clear that Figure 2A shows the cell population gated by CD8. Details should be provided. If Figure 2A represents CD8 gated cells, there are at least two or three subsets were recognized based on the fluorescence intensity (dull, medium, and very high). Since the staining pattern of CD8+ T cells is unusual (please compare with Figure 3A), the authors should provide an appropriate explanation.

             The antibodies used to generate the data in Figure 2A were anti-CD4-PE (Milteniy Biotec, #130-123-206) and anti-CD8 Percp-Cy5.5 (eBioscience, #45-0021-82). In addition, we used CD4-PerCP-Vio700 (Milteniy Biotec, #130-118-794) and CD8-PE (Milteniy Biotec, #130-123-781) in Figure 3B. In the initial experiment, we observed two CD8+ populations (Figure 2A). We repeated the experiments and observed the same flow cytometric pattern. We contacted the antibody vendor (eBioscience), believed the effect to be caused by the inclusion of NK cells or possibly degradation of the labeling dye. We excluded the possibility of NK-cell inclusion because we performed the analysis after gating for CD3+ cells. In addition, we observed the same flow cytometric analysis pattern with isolated CD8+ cells after staining with a newly provided antibody. Therefore, the detection of two CD8+ populations was likely to have been caused by the intrinsic characteristics of the antibody, and both populations in Figure 2A were considered to be CD8+ T-cells. Furthermore, the same antibody previously resulted in an identical pattern (Ge et al., 2020, Biomed. Pharmacother. 121:109626). We have used the antibody from Milteny Biotec in subsequent experiments. Regarding the gating strategy, please refer to the results of the flow cytometric analysis.

Regarding the gating strategy, please refer the flow cytometric analysis results shown above.

  1. The cytokine beads array analysis was performed using T cell culture supernatants containing CD4+ T cells that can produce Th1, Th2, and Th17 cytokines in addition to CD8+ T cells. In this regard, the authors should specifically determine these cytokine responses by CD8+ T cells.

We analyzed cytokine expression by antigen-stimulated T-cells to assess the T-cell response to immunization. To evaluate the cytotoxic T-cell response, we analyzed granzyme B expression on Tc1, a major type of cytotoxic T-cells, rather than cytokine profiles, because the aim was to determine the functions of cytotoxic T-cells after immunization.

  1. The nasal virus challenge experiments should be performed 2 months after the last nasal immunization since hACE2 KI mice showed increased levels of antigen-specific IgG and IgA antibody responses in the lungs.

In this study, a challenge experiment was conducted to determine whether the antigen-specific immune response induced by vaccination provided protective immunity against viral infection. We also evaluated whether memory cells would be maintained for 2 months after the last immunization and could respond immediately to target antigens. Although we did not conduct a challenge experiment 2 months after the last immunization as the reviewer indicated, we believe that the results support the protective efficacy of vaccination against virus infection.

  1. Since nasal immunization with HBD-PLpro19-Co1 induced significantly increased levels of antigen-specific IgG and IgA antibody responses in the respiratory tract, the authors should determine the role of each antibody isotype in the protection against viral challenge. IgG is recognized by FcγR-expressing antigen-presenting cells and can induce ADCC or phagocytosis (Bruhns et al., 2015, Immunol. Rev. 268:25). It can mediate complement activity, which may delay or inhibit virus propagation by removing virus-infected cells via CDC (Bruhns et al., 2015, Immunol. Rev. 268:25). In mice, the main IgG isotypes that mediate CDC and ADCC are IgG2a and IgG2b. As shown in Figure 1D, which is an addition to the revised manuscript, the main isotypes of PLpro19-specific IgG induced by immunization were IgG1 and IgG2b. Given that IgG1 mediates CDC, the high levels of IgG1 and IgG2b in immunized mice likely promoted the removal of virus-infected cells.

We could not analyze the subclasses of PLpro19-specific IgA because we were unable to obtain a kit for analyzing IgA subclasses. However, IgA triggers FcαRI signaling, which amplifies inflammatory responses at mucosal and nonmucosal sites via the formation of immune complexes, which orchestrate the defense against pathogens (Hansen et al., 2019, Cell. Mol. Life Sci. 76:1041). IgAs bind to PLpro proteins exposed on virus-infected cells in the mucosa and are recognized by Fcα/μR-expressing antigen-presenting cells, which may delay and interfere with virus transmission via endocytosis and/or phagocytosis (Bruhns et al., 2015, Immunol. Rev. 268:25). We have added text to lines 280–283 on page 6 accordingly.

  1. The authors should discuss the strength of using the nasal HBD-PLpro19-Co1 vaccine since the induction of humoral and cell-mediated immunity is most likely achieved by systemic vaccination strategies.

             The available SARS-CoV-2 vaccines induce a systemic immune response via intramuscular injection. Secretory IgA (sIgA) against SARS-CoV-2 and influenza virus are induced by intramuscular vaccination, albeit at low titers (Brokstad et al., 1995, J. Infect. Dis. 171:198; Sheikh-Mohamed et al., 2022, Mucosal Immunol. 15: 799). By contrast, mucosal vaccination can induce high-level sIgA expression in the mucosa (Tioni et al., 2022, NPJ Vaccines 7:85). Moreover, mucosal vaccines can protect against respiratory viruses because they induce immune responses not only at the mucosa (where viral infection begins) but also systemically (Tioni et al., 2022, NPJ Vaccines 7:85). In addition, mucosal vaccination increases the number of resident T-cells in tissues (Hassan et al., 2021, Cell Rep. Med. 2:100230). Consequently, we believe that intranasal vaccination could rapidly induce a strong defense response to respiratory viruses, especially at the respiratory mucosa. We have added the above description to lines 488–497 on page 16 in the Discussion section.

Minor Point:

  1. C57BL6 mice should be written as C57BL/6 (Figure 1 legend).

We have revised line 292 on page 7 and line 321 on page 9 accordingly.

Reviewer 2 Report

Comments and Suggestions for Authors

I do not see any major concerns with this manuscript, a mucosal vaccine is still necessary to prevent COVID-19. 

These are my minor comments

Line 42, "corona virus vaccine", use coronavirus and be specific since there are a lot or coronaviruses that cause a coronavirus disease.

Line 44-45, "However, variants of SARS-CoV-2 have mutations in S proteins, including in the RBD 44 region, which can reduce the efficacy of these vaccines", please briefly mention a couple of works that establish how much this efficiency has decreased due to new variants.

Section 2.2, specify where the his-tag is located in the construction

Section 2.2, specify the SARS-CoV-2 variant from which PLpro was based on

Section 2, details of protein expression and purification must be provided

Section 2.3, indicate how many mice were used

Use mL and not ml throughout the document

Section 3.1, include the purity of the purified protein and its yield

Discussion section, this section is presenting the results again with a little discussion. Enhance this section, are there other works using the same strategy as yours (using more conserved, less exposed sequences)? Did they obtain similar results in terms of protection?

Conclusions section, enhance this section by including key results. 

Comments on the Quality of English Language

English is fine, a few errors exist

Author Response

Reviewer #2

Comments and Suggestions for Authors

I do not see any major concerns with this manuscript, a mucosal vaccine is still necessary to prevent COVID-19.

             We thank Reviewer #2 for the positive evaluation of our manuscript and constructive suggestions to strengthen its quality. We have revised the manuscript to address the concerns. We hope that the revised manuscript has been sufficiently improved for acceptance for publication. Our responses to the specific points raised by the reviewer are provided below and indicated as grey background in the revised manuscript.

These are my minor comments

Line 42, "corona virus vaccine", use coronavirus and be specific since there are a lot or coronaviruses that cause a coronavirus disease.

We have revised line 42 on page 1 accordingly. In addition, we name particular coronaviruses thereafter.

Line 44-45, "However, variants of SARS-CoV-2 have mutations in S proteins, including in the RBD 44 region, which can reduce the efficacy of these vaccines", please briefly mention a couple of works that establish how much this efficiency has decreased due to new variants.

We have added a description to lines 45–56 on page 2 in the Introduction section.

Section 2.2, specify where the his-tag is located in the construction.

We have added a description to line 148 on page 3 as the reviewer recommended.

Section 2.2, specify the SARS-CoV-2 variant from which PLpro was based on

We have added a description to lines 137–138 on page 3.

Section 2, details of protein expression and purification must be provided.

We have added the details to lines 148–156 on pages 3–4.

Section 2.3, indicate how many mice were used

We have added the number of mice used to each figure legend.

Use mL and not ml throughout the document.

We have corrected ‘ml’ to ‘mL’.

Section 3.1, include the purity of the purified protein and its yield

We have added a description to lines 251–252 on page 5.

Discussion section, this section is presenting the results again with a little discussion. Enhance this section, are there other works using the same strategy as yours (using more conserved, less exposed sequences)? Did they obtain similar results in terms of protection?

We have added a description to lines 473–485 on page 16 in the Discussion section.

Conclusions section, enhance this section by including key results.

We have added key results to lines 554–557 on page 17 in the Conclusions section.

Reviewer 3 Report

Comments and Suggestions for Authors

The authors develop a vaccine against Sars CoV using a protein conserved in different variants. The work is interesting and is structured very well.

Minor comment:

Figure 2a: Why are there 2 or even 3 CD8 populations? Is it possible to see the dot plot even with the negative polarization?

Lane 373 and 379: why don't the authors use the same gene as a control?

Author Response

Reviewer #3

Comments and Suggestions for Authors

The authors develop a vaccine against Sars CoV using a protein conserved in different variants. The work is interesting and is structured very well.

             We thank Reviewer #3 for the positive evaluation of our manuscript and constructive suggestions to strengthen its quality. We have revised the manuscript to address the concerns. We hope that the revised manuscript has been sufficiently improved for acceptance for publication. Our responses to the specific points raised by the reviewer are provided below and indicated as grey background in the revised manuscript.

Minor comment:

Figure 2a: Why are there 2 or even 3 CD8 populations? Is it possible to see the dot plot even with the negative polarization?

             The antibodies used to generate the data in Figure 2A were anti-CD4-PE (Milteniy Biotec, #130-123-206) and anti-CD8 Percp-Cy5.5 (eBioscience, #45-0021-82). In addition, we used CD4-PerCP-Vio700 (Milteniy Biotec, #130-118-794) and CD8-PE (Milteniy Biotec, #130-123-781) in Figure 3B. In the initial experiment, we observed two CD8+ populations (Figure 2A). We repeated the experiments and observed the same flow cytometric pattern. We contacted the antibody vendor (eBioscience), believed the effect to be caused by the inclusion of NK cells or possibly degradation of the labeling dye. We excluded the possibility of NK-cell inclusion because we performed the analysis after gating for CD3+ cells. In addition, we observed the same flow cytometric analysis pattern with isolated CD8+ cells after staining with a newly provided antibody. Therefore, the detection of two CD8+ populations was likely to have been caused by the intrinsic characteristics of the antibody, and both populations in Figure 2A were considered to be CD8+ T-cells. Furthermore, the same antibody previously resulted in an identical pattern (Ge et al., 2020, Biomed. Pharmacother. 121:109626). We have used the antibody from Milteny Biotec in subsequent experiments. Regarding the gating strategy, please refer to the results of the flow cytometric analysis.

Regarding the gating strategy, please refer the flow cytometric analysis results shown above.

Lane 373 and 379: why don't the authors use the same gene as a control?

The primers used to detect the Wuhan strain of SARS-CoV-2 were for the S gene and were designed to detect other variants prior to the emergence of the Delta variant. Consequently, we were unable to detect the Delta variant using these primers in virus challenge experiments. Therefore, we prepared N gene-targeting primers suitable for detecting the Delta variant based on other studies and used them for qRT-PCR analysis.

Reviewer 4 Report

Comments and Suggestions for Authors

This manuscript describes the development of a mucosal vaccine, using the PLpro domain of nsp3 of SARS-CoV-2 combined with human beta-defensin 2, and Col. This vaccine induced higher IgA in BALF than the other groups. It also induced efficient memory T-cell responses, and could protect the mice against both the Wuhan strain and Delta variant of SARS-CoV-2 infection by inducing humoral and CTL responses. The experimental design has a certain degree of innovation, but there are still some issues that need improvement.

(1) Line 134-141, the authors expressed the antigens with an E. coli expression vector system. Why not use a eukaryotic expression system? The eukaryotic expression system is beneficial for maintaining the translation modification of viral antigen expression and maintaining its immunogenicity.

(2) Line 144, the mice were intranasal immunized once a week for 5 weeks. For a vaccine, if it takes 5 immunizations to achieve good immune protection, it cannot be considered a good vaccine.

Author Response

Reviewer #4

Comments and Suggestions for Authors

This manuscript describes the development of a mucosal vaccine, using the PLpro domain of nsp3 of SARS-CoV-2 combined with human beta-defensin 2, and Col. This vaccine induced higher IgA in BALF than the other groups. It also induced efficient memory T-cell responses, and could protect the mice against both the Wuhan strain and Delta variant of SARS-CoV-2 infection by inducing humoral and CTL responses. The experimental design has a certain degree of innovation, but there are still some issues that need improvement.

             We thank Reviewer #4 for the positive evaluation of our manuscript and constructive suggestions to strengthen its quality. We have revised the manuscript to address the concerns. We hope that the revised manuscript has been sufficiently improved for acceptance for publication. Our responses to the specific points raised by the reviewer are provided below and indicated as grey background in the revised manuscript.

(1) Line 134-141, the authors expressed the antigens with an E. coli expression vector system. Why not use a eukaryotic expression system? The eukaryotic expression system is beneficial for maintaining the translation modification of viral antigen expression and maintaining its immunogenicity.

The use of eukaryotic expression systems to express viral antigens is advantageous, but unfortunately, the one we used has low yields. Additionally, the aim was to verify the potential of PLpro as a vaccine antigen and the efficacy of a recombinant PLpro conjugate after mucosal vaccination. Therefore, we used a highly efficient E. coli expression system available in our laboratory. Although we agree with the reviewer, we contest that we obtained meaningful results using protein antigens expressed in E. coli.

(2) Line 144, the mice were intranasal immunized once a week for 5 weeks. For a vaccine, if it takes 5 immunizations to achieve good immune protection, it cannot be considered a good vaccine.

A vaccine is considered good if it elicits a sufficient immune response to the target antigen after one or two immunizations. However, it is difficult to induce a sufficient antigen-specific immune response with one or two intranasal administrations. The level of antigen-specific serum IgG is reportedly similar to the negative control following two intranasal administrations of antigen plus adjuvant, and significantly lower than that induced by subcutaneous injection (Muranishi et al., 2023, Vaccines 12:5). In addition, the level of antigen-specific serum IgG after two subcutaneous injections was similar to that induced by four intranasal administrations (Muranishi et al., 2023, Vaccines 12:5). In a previous study, we performed five intranasal administrations of recombinant antigen to evaluate the effectiveness of HBD2 as a mucosal immunization booster (Kim et al., 2020, Vaccines 8:635). The magnitude of the antigen-specific immune response was assessed after three intranasal immunizations, and no clear between-group difference was observed; therefore, five immunizations were considered appropriate for analyzing efficacy after intranasal administration of a recombinant vaccine antigen. We agree that subcutaneous or intramuscular vaccination induces a more intense systemic immune response than intranasal vaccination. However, although intranasal vaccination requires multiple immunizations, mucosal vaccines are beneficial because they induce high levels of antigen-specific antibodies and tissue-resident memory T and B cells at the site of infection (Pilapitiya et al., 2023, EBioMedicine 92:104585; Nelson et al., 2021, J. Virol. 16:e00841). Furthermore, although further studies are needed to improve the antigenicity of mucosally administered antigens, the advantages of mucosal vaccines, such as ease of administration, noninvasiveness, high patient compliance, and suitability for mass vaccination, will drive the development of intranasally administered vaccines against respiratory viruses.

Round 2

Reviewer 1 Report

Comments and Suggestions for Authors

The authors have addressed most of the concerns raised by this reviewer; however, some of the responses are inappropriate. These concerns must be addressed to improve the quality of the manuscript.

Major Points

1.    Since the duration of maintenance of protective immunity against SARS-CoV-2 is one of the major issues in vaccine development, the authors should perform the nasal virus challenge experiments 2 months after the last nasal immunization.  

2.    Mice do not have the IgA subclass like humans. 

3.    The authors speculated that FcαR-expressing cells are involved in IgA-mediated immune protection and this could be part of the role of IgA. Therefore, this statement should be included in the Discussion section. FcαR-mediated IgA protection most likely occurs after viral infection. Since secretory IgA (SIgA) antibodies are present at the mucosal surface where the virus encounters the host, the authors should also discuss the role of antigen-specific SIgA antibodies.

Minor Point:

1.    Secretory IgA should be abbreviated to SIgA.

Author Response

Reviewer #1

Comments and Suggestions for Authors

The authors have addressed most of the concerns raised by this reviewer; however, some of the responses are inappropriate. These concerns must be addressed to improve the quality of the manuscript.

             We thank Reviewer #1 for the positive evaluation of our revised manuscript and constructive suggestions to strengthen its quality. We have revised the manuscript to address the concerns. We hope that the revised manuscript has been sufficiently improved for acceptance for publication. Our responses to the specific points raised by the reviewer are provided below and indicated as pink background in the revised manuscript.

Major Points

  1. Since the duration of maintenance of protective immunity against SARS-CoV-2 is one of the major issues in vaccine development, the authors should perform the nasal virus challenge experiments 2 months after the last nasal immunization.  

             We absolutely agree that maintenance of protective immunity is an important issue in vaccine development. We also agree to the reviewer that it would be better to do a viral challenge 2 months after the last vaccination to confirm vaccine efficacy. However, it is not easily feasible to conduct additional virus challenge experiment in a short period of time because it takes a long time to obtain the necessary number of hACE2 KI mice to conduct additional challenge experiments. Also, it takes a long time to go through the required process for additional challenge experiment such as approval for animal experiments with SARS-CoV-2 and use of a biosafety level 3 facility etc. Fortunately, we found that antibody and T-cell responses were increased and maintained 2 months after the last immunization. We believe that this observation indirectly confirms that the protective immune response against SARS-CoV-2 is maintained at least 2 months after the last immunization. In addition, one of the ultimate concerns of this study is to use HBD2 and Co1 as mucosal vaccine adjuvant together with PLpro as a vaccine antigen in a laboratory level. We regret that we are not able to include the result for challenge experiment as the reviewer suggested, and will conduct further practical approach with the commercial sector in the near future.

  1. Mice do not have the IgA subclass like humans. 

             We agree to the reviewer and made an incorrect description in ‘Response to the reviewer’ in the first round revision. Additionally, it is included in this revised manuscript to line 527-528 of page 17.

  1. The authors speculated that FcαR-expressing cells are involved in IgA-mediated immune protection and this could be part of the role of IgA. Therefore, this statement should be included in the Discussion section. FcαR-mediated IgA protection most likely occurs after viral infection. Since secretory IgA (SIgA) antibodies are present at the mucosal surface where the virus encounters the host, the authors should also discuss the role of antigen-specific SIgA antibodies.

IgG is recognized by FcγR-expressing antigen-presenting cells and can induce ADCC or phagocytosis (Bruhns et al., 2015, Immunol. Rev. 268:25). It can mediate complement activity, which may delay or inhibit virus propagation by removing virus-infected cells via CDC (Bruhns et al., 2015, Immunol. Rev. 268:25). In mice, the main IgG isotypes that mediate CDC and ADCC are IgG2a and IgG2b. As shown in Figure 1D, the main isotypes of PLpro19-specific IgG induced by immunization were IgG1 and IgG2b. Given the possibility that IgG1 can mediate CDC (Vukovic et al., 2023, Cancer Res. Commun. 3:109), the high levels of IgG1 and IgG2b detected in immunized mice likely promoted the removal of virus-infected cells. In the case of mouse IgA, unlike humans, has only one IgA isoform and do not have a functional homologue with FcαRI (Hansen et al., 2019, Cell. Mol. Life Sci. 76:1041). However, IgA-mediated effector functions via other IgA receptors including Fcα/μR (CD351) and transferrin R (CD71) whose expression has been reported in mice, are conceivable (Bruhns et al., 2015, Immunol. Rev. 268:25; Decot, V et al., 2005, J. Immunol. 174: 628). IgAs bind to PLpro proteins exposed on virus-infected cells in the mucosa and are recognized by FcR-expressing antigen-presenting cells, which may delay and interfere with virus transmission via endocytosis and/or phagocytosis (Bruhns et al., 2015, Immunol. Rev. 268:25). We believe that neutralization through direct binding of SIgA is difficult because PLpro-specific SIgA cannot bind to viral particles. However, increased SIgA can induce immune exclusion, making it difficult to access the mucus membrane by capturing the pathogen in the mucus layer (Li Y et al., 2020, Biomed Res. Int. 2020). Therefore, it is thought that increased SIgA has the potential to reduce viral infection and transmission.

             We have added the above description to lines 520-537 on page 17 in the Discussion section.

Minor Point:

  1. Secretory IgA should be abbreviated to SIgA.

             We have corrected ‘sIgA’ to ‘SIgA’ in line 489 and 491 on page 16.

Reviewer 4 Report

Comments and Suggestions for Authors

The authors have made appropriate revisions and clarifications to the manuscript.

Round 3

Reviewer 1 Report

Comments and Suggestions for Authors

None